# Challenges of Providing Home Care for a Family Member with Serious Chronic Mental Illness: A Qualitative Enquiry

**DOI:** 10.3390/ijerph17228440

**Published:** 2020-11-14

**Authors:** Kebogile Elizabeth Mokwena, Amukelane Ngoveni

**Affiliations:** 1Department of Public Health, Sefako Makgatho Health Sciences University, Pretoria 0204, South Africa; 2Weskoppies Psychiatric Hospital, Pretoria 0200, South Africa; ngovear2@gmail.com

**Keywords:** mental disorders, South African Mental Health Act of 2002, families, caregivers, homecare

## Abstract

The South African Mental Health Act of 2002 advocates the de-institutionalization of treatment of patients with mental disorders, so that the mental health care users or patients are treated in their communities. Although this approach is often used to discharge patients from hospital, no feasibility assessments are conducted to ascertain adequate care for these patients. The objective of the study was to explore the experiences of family members who provide home care for patients with serious mental disorders. A qualitative explorative design was used to interview 20 primary caregivers whose family members were readmitted to a public psychiatric hospital in Pretoria. Data were analysed using NVivo version 11. The findings are that caring for patients with serious mental illness at home is difficult, sometimes unbearable, because the families have to deal with violence perpetrated by the patients, safety concerns, financial difficulties and emotional turmoil, and wish that the patients would be kept in institutions. The absence of required skills and resources to care for the mentally ill at home exposes the patients and their families to emotional, financial and social difficulties, and results in unfavourable outcomes for both the patients and their families.

## 1. Introduction

Mental disorders have emerged as a growing and significant public health challenge, with its global burden estimated to account for significant proportion of both years lived with disability (YLDs) and disability-adjusted life-years (DALYs). There are even views that the true burden of mental illness is often underestimated, and that its true burden is much higher than currently estimated [1]. Globally, resources for the treatment of mental disorders are inadequate [2], and this is much worse in low and middle-income countries [3,4]. Moreover, even within a country, mental health resources are not evenly distributed within communities, which results in communities with high rates of socioeconomic deprivation having the highest need for mental health care services, but the lowest access to it [2,5], with resultant overall poor population mental health.

Compared to other African countries, mental illness is reported to be higher in South Africa [6], which indicates a need to increase both the human and financial resources required for a positive impact on health outcomes of patients treated for a variety of mental illnesses. However, with both the prevalence and severity of mental disorders reported to be on the increase, the resources allocated to the treatment of mental disorders have not increased proportionally in South Africa [7,8].

Literature reports on an association between low socio-economic status and increased prevalence and severity of mental disorders for both adults [9] and children [10], which partly explains the high prevalence of mental disorders among poor people and in developing countries. This then creates a situation of pockets of higher prevalence of mental illness and associated lower access to health services in certain communities. It is often within such communities that people are vulnerable to developing mental disorders and their family members are expected to provide care though they are not well equipped and enabled to provide it.

Home care for patients with chronic mental disorders continues to attract attention in health care research and service provision. According to the World Health Organization guidelines for the management of adults with severe mental disorders [11], equitable access to comprehensive health services is key to providing a service that enables the patients to access the same quality of care as the general population. However, many people with mental disorders often lack access to health services or receive poor quality care, as characterized by promotion and prevention, screening and treatment. Other frameworks on the requirements for a successful homecare program for people with mental disorders requires components of psychosocial, physiological, medical support and social services, which need a deep level of organized contact with the families of the patients to be able to provide the patients with problem solving skills [12]. On the other hand, a dependable hospital information technology is essential to supporting families who practice home care for patients with mental disorders [13]. Without the proper support and treatment components, homecare was found to contribute to a higher prevalence of depressive symptoms in Europe [14].

Section 4 of the South African Mental Health Care Act (SAMHCA) No. 17 of 2002 advocates community-based care, treatment and rehabilitation services of mental health care users, while Section 8 states that such care should be provided in a way that facilitates community care of mental health care users [15]. Together, these sections of the Act have resulted in a practice where psychiatric hospitals discharge patients with mental disorders to be cared for by their family members. However, this transition requires skills, funds and other resources, which are not availed to the family or community to which the patients are discharged [2], more so among poor people. There are, therefore, no specific mechanisms, programs or processes for the preparation, support, skill transfers or availing of additional resources for families who undertake the responsibility of caring for their mentally ill relatives. In the context of such support and resources not being provided, and the paucity of published studies in this area in South Africa, the purpose of this study was to explore the experiences of families who care for their mentally ill relatives at home.

Although medication is viewed by some to be critical for the treatment of serious mental disorders, it brings about challenges of adherence for such patients as they need quality supervision to take their medication as required [16]. On the contrary, there is some contention for the use of medication for this class of diseases, with some scholars attributing the worsening of the mental condition to the very medication that is meant to improve it, and that such medications actually contribute to the current epidemic of mental disorders [17]. However, even among those that accept the value of medication as the main component of treating mental disorders, the treatment of chronic mental illness requires not only the medication to manage the symptoms but also approaches that will acknowledge and attend to a range of aspects of the patient’s life, including psychosocial support [18]. However, when patients are discharged from facilities as per the principles of the South African Mental Health Care Act (SAMHCA) No. 17 of 2002 [15], none of such support is provided and at best, the family is given medication only, which is inadequate for the patient. This then renders the patient vulnerable for relapse, especially with a family overwhelmed by its limitations to provide the required care and support.

The social and physical health impact of mental disorders is not limited to the patient, but also extends to the family, as mental distress has been identified among family members of psychiatric patients [19,20]. Children of parents with mental illness are at risk of psychiatric and behavioural problems which include poor academic performance, inability to sustain committed relationships, alcohol and drug problems as well as problems with the law [14]. Moreover, families of patients with mental disorders also deal with financial strains, the disruption of domestic routine, constraints to social and leisure time, physical violence, damage to property presenting challenges to the families as well as the stigma that is commonly directed at people with mental illness [21].

### Objective

The objective of this study was to explore the experiences of providing homecare for a family member with serious mental disorders.

## 2. Study Methods

### 2.1. Study Design

An explorative qualitative design, using an in-depth interview guide, was used to collect data.

### 2.2. Study Population and Sample

The population consisted of family members who assumed the responsibilities of providing homecare for their relatives who had serious mental illness. A sample was selected from those whose relatives were readmitted to Weskoppies psychiatric hospital in Pretoria, South Africa. The participants were both males and females who were 18 years of age or older and were primary caregivers of patients diagnosed with chronic mental disorders during the period of data collection.

### 2.3. Study Setting

The study was conducted at Weskoppies Hospital, which is a referral public psychiatric hospital in Pretoria, Gauteng. The hospital is a teaching/training hospital by two Universities in Gauteng, and has a bed capacity of 850.

### 2.4. Sampling and Sample Size

A purposive sampling technique was used, which intentionally identified primary caregivers who were considered to be best suited to provide information to meet the objective of the study, were willing to participate in the study and were available for data collection/interview. The sample size was determined by data saturation, which is when new additional interviews no longer provide new information. Data saturation was reached after twenty (20) interviews were conducted.

### 2.5. Inclusion Criteria

Primary caregivers whose relatives were previously admitted to hospital, were discharged to be cared for at home and were admitted to hospital again at the time of data collection were included. Serious mental illness was limited to bipolar, schizophrenia and major depression as these chronic disorders were identified to be common in developing countries [21,22] including South Africa [23]. These inclusion criteria, therefore, supported the objective of the current study.

### 2.6. Exclusion Criteria

Patients who were diagnosed with mental disorders other than bipolar, major depression or schizophrenia were excluded, as were those who were not previously discharged from hospital and cared for at home. Relatives who were visiting the patient but were not primary caregivers were excluded.

### 2.7. Recruitment of Study Participants

Relatives of the patients who were readmitted to hospital were recruited by the researcher on an individual basis when they came to visit their relative who was being treated for chronic mental disorder. The purpose of the study was briefly explained and it was established if the potential participant was a primary caregiver or just shared a home with the patient when they were out of hospital and were being cared for in the community. The appropriate potential participant was then requested to participate in the study and an appointment was made for the interview.

### 2.8. Data Collection

Data collection through in-depth interviews was conducted in a room within the ward or a room at the hospital central admission area, where privacy and confidentiality could be obtained. The interviews were conducted in English, Setswana or Xitsonga, as determined by the preference of each participant. Demographic data were collected first, which was followed by the in-depth interview, which was digitally recorded. Interviews continued over a period of 6 weeks until data saturation was reached, when additional interviews no longer provided new information.

### 2.9. Data Analysis

The demographic data were analysed quantitatively and descriptively. The digital and qualitative audio recordings were transcribed, translated into English, typed into Word and transported into Nvivo version 11 (QSR International, Melbourne, Australia) for analysis. Both authors conducted the initial thematic analysis by reading the first transcript several times, and identified phrases and/or sentences that related to challenges of experiences of providing homecare to patients who have chronic mental disorders. Phrases and/or sentences that reflected the same view were copied verbatim and grouped together under a code or theme and named in line with what the phrases/sentences collectively reflected. A codebook was created from the first few transcripts, with several codes and definitions of each code, and these codes were applied to all the transcripts. The codebook was refined as more transcripts were coded. The final codebook, with related definitions, was worked on until agreed upon by both authors. The verbatim phrases/sentences were used to support the theme under which they were coded during the writing of the narrative.

### 2.10. Ethical Considerations

Ethical approval for the study was obtained from the Medunsa Research and Ethics Committee (MREC/H/365/2014:PG) before data collection commenced. Permission to conduct the study was obtained from the management of the hospital. Each participant provided informed consent before they were interviewed.

## 3. Findings

### 3.1. Characteristics of the Participants and Their Relatives

Twenty (20) participants, two males and eighteen females, who were visiting twenty patients, were interviewed. Most (80%, *n* = 16) were Black Africans, two were White, one was Indian and one Coloured. The ages of the participants ranged between 18 and 80, and the ages of the patients ranged between 13 and 77 years. The diagnosis of the patients consisted of equal proportion of schizophrenia and psychosis (35%, *n* = 7 each) followed by bipolar mood disorder (15%, *n* = 3) depression (10%, *n* = 2) and attention deficit and hyperactivity (5%, *n* = 1). Table 1 shows the summary of the socio-demographic variables of both the mental health care user/patient and the participants/caregivers.

### 3.2. Findings of the Qualitative Data

After a process of refining the original codes, dismantling and merging some, the following are the final themes that portray the findings and, therefore, meet the objective of the study.

### 3.3. Obligation to Care for a Mentally Ill Relative

Family members view and accept the caring of relatives with mental disorders as an obligation, or some moral duty which needs to be carried out despite the difficulties they may experience. This obligation implies that in their view, they have no other choice but to provide care, and this becomes an emotional burden, which they cannot escape from.

*“When my mother passed away, …there was no one to look after him. My sister and I therefore decided to share the custodianship of him” (a 45-year old female, caring for her brother), and “I have to suffer, he is my son. Who else can take someone’s burden?”* (a 50-year old female, caring for her son).

### 3.4. Violent and Aggressive Behaviour

The violence and aggression were both real and threatening, which occur as a manifestation of the mental disorder. Although some of the aggression is directed at the caregivers and immediate family members, some of it is directed to members of the community.

*“If I have to open up to you, my son does assault me physically. He once slapped me on the face when I told him about his father’s absence in my life”.* (50-year old female who cares for her son) and *“One day during evening time he took a baby girl of his fellow church member and ran with her to the nearest river where he tried to drown her.”* (a 50-year old female caring for her husband).

Sometimes the aggressive behaviour by the person with mental illness is reciprocated by the caregiver, as expressed by a 41-year old male who was caring for his brother:


*“I don’t think that it is always good to beat them, but the way we are stressed up leads to that situation”.*


### 3.5. Concerns about the Safety of the Patient

The participants raised concerns about the safety of their relative who has mental illness, because they often act in ways that may be viewed as provocative to people who do not understand them. They were also concerned that the mentally ill may hurt themselves or even get lost.

*“If he feels that he was not treated fairly, he goes out on the yard and looks for some place to hide. My yard is very big and full of scrap and many dangerous objects. He exposes himself to high risk of injury. Snakes, falling objects and sharp objects are a real risk”* (25-year old male, caring for his brother).

The safety concerns include suicide threats, as depicted by

*“Sometimes she becomes so sad that she threatens to kill herself. She has already taken overdose of her psychiatric treatment and spent few days in a general hospital. She sometimes threatens to hang herself, citing that life is useless and thus what is the point of living.”* (A 61-year old female, caring for her daughter) and *“The first time she took an overdose of tablets. She was helped in Eugene Mare hospital. Counselling and everything were done. I thought it was over, she repeated the overdose of medication. I took her to the very same hospital and they helped her. They then referred me to Weskoppies”* (49-year-old male, caring for his daughter).

### 3.6. Financial Difficulties Experienced by the Family

Financial difficulties occur because of high unemployment rates among the patients and caregivers, as well as costs related to caring of the mentally ill. Because mentally ill patients have poor prospects of being employed, many of them depend on family members for their up-keeping, which increases financial vulnerability to the whole family. Moreover, the needs of the mentally ill places an additional financial burden on the often financially struggling family. This view was expressed as “*I pay close to five hundred rand (R500-00) a return trip. When I arrive here, I have to make sure that I have something to give to him. On the other hand, I am also sick with my limb. I have to see my physiotherapist at least once every three months. It all involves money*” (A 41-year old female).

### 3.7. Emotional and Psychological Impact

Almost all the participants in the study reported to have some emotional and/or psychological distress as they find it difficult to live with a loved one who is mentally ill, and provide the care that the patient needs. This was expressed as “*I sometimes become very exhausted emotionally and reach a point where I do not know what to do….I am living in a continual stressful situation. I always have severe headaches, I do not even know what to do’’,* (41-year old female, caring for her son) and “*It really causes some emotional disturbance. In other words, we are living in sorrow that never ends*” (18-year old male, caring for his brother). A specific source of emotional distress results from some community members doubting whether the patient is really mentally sick, and was articulated as follows: *“Some of them, including my neighbours, even think that maybe I support what he (son) is doing. Some of them even say that we have never seen him pick up papers, which is a good indication of mental condition”* (43-year old female, caring for her son).

### 3.8. Societal Stigma Directed to the Patient and the Family

Societal stigma was reported to be directed to both the patient and the family and was expressed as being viewed in a negative way because of the existence of mental illness. This extends to discrimination where some members of the community treated both the family and the patient in a negative way, “*Some of them, including my neighbours, do not greet me because of my son’s behaviour. They even think that maybe I support what he is doing.*” (A 43-year old female caring for her son). An effort to address the stigma included summoning the community for understanding and help, and was articulated as follows:

*“I even addressed the community ward meetings about his mental state so as to make every one aware and to help support us in our quest to get him well”* (66-year old male, caring for his son).

### 3.9. The Expressed Need for Continuation of Institutional Care for the Mentally Ill

The need for the continuation of institutionalized treatment of the mentally ill emanated from their helplessness and inability to provide for the needs of the mentally ill—“*I think that he needs some specialised care which can be offered by professional bodies including government hospitals”* (45-year old male, caring for his brother)—and their view is that trained and equipped professionals should be providing this care. The need for institutional care was also driven by the rejection of the patient and the family by the community—“*I think if my brother cannot get cured, the government should find him a permanent home*” (48-year male, caring for his brother) and “*Yes that is the reason we have taken him for admission to this hospital. We reached a point where we felt that our lives are in danger more especially our children’s*.” (a 46-year old female, caring for her brother).

## 4. Discussion

With the global increase in the prevalence of mental disorders, the treatment and management protocol for the mentally ill is often in the spotlight, and this study highlights the impact of the treatment protocol on the overall outcome of not only the patient but also the family. The findings of this study suggest that the mental disorder of many of the patients is not well managed and controlled, and their being cared for at home is not conducive to a favourable outcome.

The sample in this study consisted of mostly Black people, whose high unemployment rates have been previously reported [24] and, therefore, remain economically and socially disadvantaged, which explains the common financial concerns that were reported by the participants. The situation of high unemployment rates among Black South Africans is made more complex by reported high unemployment rates among people with mental disorders. The complexities of inadequate resources for prevention and treatment of mental illness in communities with low socio-economic status increases the costs of such resources and the resultant financial burden of affected families [25,26,27,28]. The situation in which the participants of the study find themselves creates a cycle of poverty and vulnerability to mental disorders such as depression and anxiety [29].

On the other hand, unemployment among patients who have mental disorders weakens the employment protection systems because the mentally ill find it difficult to find or to keep employment [30]. Being employed does not only improve the prognosis of the mentally sick but has other benefits which include fostering of self-esteem, provision of coping strategies for psychiatric symptoms and facilitating the process of recovery [31], and thus an improvement in the overall mental health status. Moreover, unemployment has been reported to have a significant positive relationship with aggressive behaviour among people with mental disorders [32]. This suggests that the blanket application of discharging mentally ill patients to be cared for at home impacts more negatively among the poor.

Although the expressed moral obligation to care may benefit the patient in the short term, extended care results in the caregivers feeling frustrated and helpless to conduct the required caring responsibilities as the families become overwhelmed by the role, and are consumed by the difficult situation [33,34]. The obligation may thus turn to anger and animosity towards the mentally ill, which may result in various forms of abuse.

Although the family can be a key resource in the care of patients with mental illness, if the family is poor, either monetarily or because of lack of other support resources, their ability to care is substantially reduced, which results in the difficulty of living with, and caring for, their mentally ill relatives being burdensome. To enable families to care for their mentally ill relatives, they need specific empowerment programs which may include the outcome of the patient’s disease; management of the patient’s symptoms; as well as assistance with developing and reinforcing social networks [[26] acquisition of mental health literacy [35,36], and the ability to prevent relapse [37]. Moreover, they need to be trained on how to assess if the mental health condition gets out of control and what actions to take. The participants of this study did not have any of these elements of support.

Aggressive and violent behaviour, which was a recurring theme reported, seems to be a frequent experience reported by carers, and is reported to be a key challenge among people with mental illness [38,39]. The management of this aggression requires mental health practitioners with specific skills [32], skills which the caregivers do not have, nor are they provided with any form of support as their relatives are discharged home. The aggression and violent behaviours indicate that the patients’ mental disorder symptoms are not being adequately managed when the patients are being cared for at home, and this puts the well-being of both the patients and carers at emotional and physical risk. Moreover, there is lack of any interventions to assist the families in coping with their caring responsibilities and challenges, which results in emotional burden and may lead to emotional and psychological burnout and exhaustion.

The experienced societal stigma reported by the participants is similar to findings of other studies, and because mental illness often impairs a person’s capacity to perceive and to act in good judgment, such actions and attitudes from members of society may aggravate the undesirable actions of the mentally ill, which will further create animosity between him/her and other members of the family and community, and thus increase the stigma. The existence of the stigma disables the person’s integration into the community and fails to promote a cohesive level of family life. The general population, therefore, requires education to prepare them to support the person with the mental disorder and their family [33]. None of these conditions are met by the current practice of discharging patients home to be cared for by family members.

## 5. Conclusions

The South African Mental Health Act of 2002 promotes the integration of respect and human dignity into the treatment of mentally ill patients, but the findings of this study suggest otherwise [15]. The current system, in which patients are discharged home to be cared for by their relatives without due considerations for either the patient or family’s wellbeing, is neither beneficial nor ethical for the patients and their families. Health literature identifies conditions for ongoing care needed for patients who are cared for at home, and these include psychosocial, physiological, medical support and social services [12], none of which were provided for the participants in this study. Most of these patients are only given medication, which is not adequate for the promotion of mental health which they so need. In its current form, the practice, therefore, contributes to the abuse of human rights for the mentally ill, as well as their families and communities, who are exposed to situations which can develop into long-term poor mental health.

i.That community-based outreach teams, which are based in primary health clinics and offer basic extended services to patients in communities, assist family members who provide homecare for discharged patients to provide some support and referrals according to their needs.ii.Carefully planned follow-up schedules should be developed to enable the continuous assessment of discharged patients, which will allow for timely appropriate decision making to support the wellbeing of such patients and their families.

## Figures and Tables

**Table 1 ijerph-17-08440-t001:** Demographics of the patients and their caregivers.

	Mental Health Care User		Participants/Caregivers	
Race	Frequency	Percentage	Frequency	Percentage
African	16	80	16	80
Indian	1	5	1	5
Coloured	1	5	1	5
White	2	10	2	10
**Marital status**	**Frequency**	**Percentage**	**Frequency**	**Percentage**
Single	18	90	4	20
Married	1	5	11	55
Divorced	0	0	4	20
Widowed	1	5	1	5
**Gender**	**Frequency**	**Percentage**	**Frequency**	**percentage**
Males	12	60	8	40
Females	8	40	12	60
**Age group**	**Frequency**	**Percentage**	**Frequency**	**percentage**
18–19	2	10	1	5
20–39	12	60	3	15
40–59	4	20	13	65
60–79	2	10	3	15
**Employment status**	**Frequency**	**Percentage**	**Frequency**	**percentage**
Employed	1	5	8	40
Unemployed	19	95	12	60
**Religion**	**Frequency**	**Percentage**	**Frequency**	**percentage**
Christianity	15	75	15	75
Islam	1	5	1	5
No religion	1	5	1	5
African	3	15	3	15
**Level of education**	**Frequency**	**Percentage**	**Frequency**	**percentage**
No formal education	4	20	0	0
Primary education	8	40	9	45
Secondary education	3	15	10	50
Tertiary education	5	25	1	5
**Period of mental illness in years**	**Frequency**	**Percentage**		
0–10	10	50		
11–20	4	20		
21–30	2	10		
31–40	3	15		
>40	1	5		
**Relationship of the participant to the patient**	**Frequency**	**Percentage**		
Parent	7	35		
Siblings	9	45		
Spouse	1	5		
Others	3	15		

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
