# Peer review of "Challenges of Providing Home Care for a Family Member with Serious Chronic Mental Illness: A Qualitative Enquiry"

_ijerph, 2020, doi:10.3390/ijerph17228440_

Round 1

Reviewer 1 Report

1.Introduction

After reading the introduction, I consider that it is necessary to introduce a theoretical-practical review of the experience of other countries regarding the care of patients with chronic mental illness (relatives, patients and professionals) in order to introduce the subject under study

3.Findings

In point 3.1. I consider that the variable of economic income level should have been reflected

I believe that it is necessary to introduce a results section in which the methodology used to categorize the responses of the relatives and to reach the conclusions is described. Since only the categories are described and an example is offered (opinions of relatives).

But it is not explained how the categories of the points are interpreted (3.3 to 3.10)

I believe that it is necessary to introduce a section of results in which the frequencies of the categories collected from relatives are shown

4.Discussion

I believe that the statement collected in lines 218-219 cannot be made based on the data presented in the study, since this statement is made based on the vision of the family members in a situation in which they only have medication as a resource . I consider that the wording of the discussion is not adequate, since its wording is not correct, it omits relevant information (the family members have no recourse other than medication).

Therefore the statement is biased, I think it should be better elaborated.

5.Conclusion

I consider that the conclusion that is gathered in lines 271-274 "The current ... families"

It must be written correctly, specifying that the current system is based on the administration of medication and no other resources are offered (day center, home hospitalization, multidisciplinary team ...).

Author Response

Please find attached my responses to the points you raised

Kind regards

Reviewer 2 Report

Thanks for inviting me to review this manuscript. This manuscript is well written and high readability. I have a few comments on the manuscript as below:

  1. Introduction

Line 38-39:  Please specify between low socio-economic status and [the issue of] mental disorders.

Line 93: Please justify the inclusion criteria (e.g. the types of serious mental illness) and how it aligned with the research question.

  1. Study methods

Please add a few sentences to describe how the quotes defined and to mention the relationship between the quotes and the codes to improve the transparency of the process.

  1. Findings

Please provide a visual display of coding output of Nvivo.

  1. Discussion

The participants of this study consist of several types of clinical conditions (bipolar, schizophrenia and major depression). The author may consider to add a few discussions on the specificity and generalizability of the findings of this study.

  1. Conclusion and recommendation

Line 282-283: The presentation of idea in this point is unclear. Please rephrase.

Author Response

Please find an attachment which responds to the matters raised in your review

Kind regards 

Reviewer 3 Report

I enjoyed reading this paper which addresses a gap in the literature regarding psychosocial support for families caring for relatives with long-term mental health problems in the South African context. I suggest a few points to help strengthen the paper.

In the background section:

The value of medication in symptom management is presented uncritically. There may be room at this point to note that the efficacy of psychotropic medication is a matter of contention for some scholars and many patients (eg Whitaker, Moncreiff etc).

I was surprised to see no explicit mention of ethnicity or ethnic disadvantage in the presentation of background/context or in the discussion of findings. The intersections between ethnicity, unemployment and social disadvantage in contemporary South Africa could be more explicitly highlighted. The categories of ethnicity referred to in the findings require further attention and/or justification. Are these in standard usage in South Africa (census terms or routinely used in services?). I suggest the notion of 'coloured' is quite dated and potentially offensive; in any event, I am not sure what it means here. Similarly, there is a somewhat jarring use of the term 'blacks' in the paper which should be taken out eg replace with 'black South Africans'.

Methodology:

The reference to purposive and convenience sampling is somewhat confusing; perhaps the purposive aspects could be succinctly elaborated on.

Similarly, the approach to data analysis lacks detail and could be referenced to a supporting citation.

Findings:

The titles of each theme could be shortened. There are some problems with the table summarising the themes, for example incomplete sentence accounting for theme 7. I am not sure there is a need for this table and it could be deleted.

Statements such as 'this theme means' could be replaced by alternatives such as 'This theme describes' or 'This theme accounts for ...'

The presentation of minor themes seems at odds with my understanding of qualitative research - a theme is a theme, regardless of how many of the participants gave voice to it. I suggest presenting these as full themes or, where appropriate, incorporating them into one of the other themes e.g. there could be a theme of violence which accounts for this generally.

The discussion is good and supports conclusions and recommendations.

Author Response

Please find an attachment with responses  to matter raised in the review of the manuscript

Kind regards

Round 2

Reviewer 3 Report

Thank you for satisfactorily attending to review feedback. The paper merits publication.